# Connections Between Cellular Senescence and Alzheimer’s Disease—A Narrative Review

**DOI:** 10.3390/ijms26178638

**Published:** 2025-09-05

**Authors:** Julia Kuźniar, Patrycja Kozubek, Magdalena Czaja, Hanna Sitka, Urszula Kochman, Jerzy Leszek

**Affiliations:** 1Student Scientific Group of Psychiatry, Faculty of Medicine, Wroclaw Medical University, 50-367 Wroclaw, Poland; patrycjakozubek99@gmail.com (P.K.); magdalena.czaja47@gmail.com (M.C.); sitka.hanna@gmail.com (H.S.); urszulakochman@gmail.com (U.K.); 2Department of Psychiatry, Faculty of Medicine, Wroclaw Medical University, 50-367 Wroclaw, Poland

**Keywords:** Alzheimer’s disease, cellular senescence, aging, senotherapies, SASP, mitophagy, senescent microglia, senescent astrocytes, senescent neurons

## Abstract

Alzheimer’s disease, a neurodegenerative brain disorder leading to the progressive decline in cognitive functions, is the most common type of dementia. The main risk factor for its development is aging. Recent studies indicate that cellular senescence mechanisms are among the major factors in a heterogeneous aging process. Cellular senescence is characterized by a permanent proliferative arrest. Many factors might initiate senescence, for example, damage of DNA, shortening of telomeres, dysfunction of mitochondria, and oncogene activation. These processes lead to alterations in the morphology and function of senescent cells. Research is still ongoing to identify one universal marker that could detect senescent cells and distinguish them from other non-proliferating cells. Those cells are involved in age-related pathologies through many heterogeneous processes, including secretion of pro-inflammatory senescence-associated secretory phenotype factors, which affect the brain differently. Alzheimer’s disease is an example of a neurodegenerative condition connected to cellular senescence. Senescent cells have been demonstrated to accumulate near Aβ plaques and neurofibrillary tangles. In this review, the multifactorial connection between Alzheimer’s disease and cellular senescence is discussed, including topics such as senescence of astrocytes, defective mitochondria, dysregulation of cellular autophagy, and the role of senescent microglia.

## 1. Introduction

Alzheimer’s disease represents the leading cause of dementia around the world. Research indicates that approximately 50 million individuals are affected. Due to global population aging and increased longevity, this number is estimated to triple by 2050 [1,2]. The main neurological factors underlying AD pathology are the formation of amyloid-beta plaques and neurofibrillary tangles consisting of hyperphosphorylated tau, as well as gliosis, degeneration of cholinergic neurons, and synaptic loss [1,2,3]. Some initial symptoms of AD can occur years before the disorder is indicated. Progression of the disease leads to disorientation, confusion, aggression, and delusions [3]. Although the changes in the central nervous system associated with cognitive decline are thought to occur years prior to the symptom onset [4]. The main problem with treating the disease is that the available therapies remain limited and focused on symptom’ relief instead of eliminating the causes [2]. Recent research points to aging as a major factor contributing to the late-onset Alzheimer’s disease (LOAD). LOAD is estimated to cause more than 95% of AD cases [5]. The aging process entails a continuous biological decline in functionality and structure of tissues and organs over time, characterized by the reduced functionality of tissues and organs over time, thereby increasing the predisposition to a spectrum of age-related pathologies, including neurodegenerative diseases. Therefore, substantial attention has been directed toward mechanisms involved in processes of cellular senescence, which are indicated as pivotal determinants to the development of AD [6]. Under persistent stress during the aging process, postmitotic and proliferative cells may entail a condition of chronic cellular senescence (CS) [7]. Regardless of the organism’s age, cells undergo senescence in response to a multitude of signals, including those independent of telomere shortening. As a result, senescent cells are present throughout life—from embryogenesis, where they participate in tissue development, to adulthood, where they play roles in limiting the proliferation of damaged cells, promoting tissue repair, and suppressing [8]. Senescent cells are distinguished by various features such as loss of proliferation ability, resistance to apoptosis, modification of metabolic functions, and secretion of active molecules (SASP) [5]. Release of SASP factors exacerbates mitochondrial disability and triggers oxidative stress in cells [9]. Senescent cells, including neurons and microglia have been found in the CNS of patients and animals [10]. Removing these cells from the mouse brain improves β-amyloid and tau protein neuropathology [10]. However, the evidence of cell senescence in AD pathology remains insufficient. Some studies have demonstrated that neural and endothelial cells may express senescence-associated factors, such as SA-β-gal, lipofuscin, p16, p53, and p21 [5].

This article explores potential links between cellular senescence and Alzheimer’s disease. We summarize recent advances in this field and emphasize the importance of targeting cellular senescence in future AD therapies. Bibliographic research was undertaken in August 2025. The articles were selected using PubMed search, employing keywords related to cellular senescence in AD. A total of 87 articles were incorporated for the analysis. In order to include all relevant studies, the search did not impose limitations on publication type or study design.

## 2. Alzheimer’s Disease—The Epidemiology, Etiopathology, Symptoms, and Treatment

Alzheimer’s disease is one of the major global healthcare concerns. It is reported that more than 50 million individuals around the world suffer from dementia, and the prevalence is estimated to reach 152 million patients worldwide [11]. The number of AD patients in the US may rise from 5.8 million to 13.8 million in 25 years [11]. In 2020, the total healthcare expenditures associated with Alzheimer’s disease and related dementias (ADRDs) in the United States were estimated at $196 billion [12]. In 2019, 121,499 deaths were listed as being due to AD [13]. Various main hypotheses are under consideration regarding AD onset and progression. Extracellular Aβ plaques are composed in various regions of the CNS, leading to neuronal death and impaired synaptic plasticity. The *APOE4* variant and mutations in *APP*, *presenilin 1*, and *presenilin 2* genes contribute to the aggregation of beta-amyloid [14]. Studies indicate that the accumulation of pathological Aβ, generated through the sequential cleavage of APP by β- and γ-secretase enzymes, constitutes the primary mechanism underlying the disruption of Aβ homeostasis between its production and clearance [1]. In addition, intracellular neurofibrillary tangles are formed by hyperphosphorylated tau. Aβ plaques deregulate p35 (the stimulator of CDK5—cyclin-dependent kinase 5). Excessive cytosolic calcium induces cleavage of p35 into p25, subsequently activating cyclin-dependent kinase-5 (CDK5) and resulting in tau hyperphosphorylation [14]. This process is driven by mutations in *tau* genes or dysregulation of kinases, including CDK5, Fyn kinase, and glycogen synthase kinase-3β (GSK3β). The accumulation of hyperphosphorylated tau disrupts synaptic transmission, axonal transport, and signal transduction [14]. Moreover, AD is characterized by synaptic dysfunction in the hippocampus which is linked with an increased glutamatergic transmission and responsiveness of pyramidal neurons. It causes a decrease in GABAergic neurotransmission inhibitors and in increased efficiency of glutamate [15]. Furthermore, persistent stimulation of microglia induces loss of Aβ removal capacity; nevertheless, the ability to secrete pro-inflammatory cytokines is preserved. This leads to an imbalance between pro- and anti-inflammatory mechanisms [14,16]. In addition, research indicates that several additional factors are associated with AD pathogenesis such as gut microbiota disturbances, oxidative stress influence, *HSV1* and *Chlamydia* infections, *HERVs*, autophagy, and many more [14]. Scientific studies demonstrate that cellular senescence contributes to the onset and progression of AD. We have tried to summarize this link in subsequent parts of the article. AD varies from the preclinical phase to mild cognitive impairment (MCI), and finally, patients suffer from severe dementia [17]. Preclinical AD is an asymptomatic stage, in which patients show evidence of AD pathology, but no cognitive decline is observed. The duration of the phase differs between people and usually lasts around 6–10 years; however, it may remain up to 15–20 years [17,18]. The risk of progression to MCI depends on age, sex, and ApoE status. However, not all patients will progress to MCI or AD dementia [17]. MCI is a symptomatic phase involving evidence of impairment in cognitive domains, though individuals retain independence in functional abilities, social, and occupational activities. Patients mainly suffer from short-term memory problems [17,19]. Around 35–85% of patients present one or more neuropsychiatric symptoms, such as apathy, depression, or anxiety [19]. The AD dementia phase, the late stage of the disease, is characterized by severe cognitive deficits affecting social and behavioral functioning. The disease results in a complete loss of independence due to severe memory impairment and impaired executive function. Likewise, neuropsychiatric symptoms are common [17,19]. Despite significant progress in understanding of pathological links in AD, therapy methods are still insufficient due to incomplete understanding of the underlying pathophysiological mechanisms of AD. Currently available drugs primarily focus on alleviating the symptoms. Therapy mainly relies on modulation of cholinergic and glutamatergic neurotransmission [20]. The drugs approved in the treatment include donepezil, rivastigmine, galantamine, and memantine. Nonetheless, these drugs do not prevent neuronal loss nor exacerbation of cognitive functions [20,21]. As a result, the development of therapies focused on eliminating the causes and disease-alleviating factors has become a healthcare priority [21]. In recent years, researchers have been investigating the first disease-modifying drug called Lecanemab (humanized IgG1 monoclonal antibody) with affinity to β-amyloid components. Consequently, the drug decreases β-amyloid plaque accumulation [22,23]. Preclinical research has demonstrated its capability to penetrate the blood–brain barrier and facilitate the clearance of Aβ plaques. Furthermore, clinical trials have provided evidence supporting its efficacy in patients with mild to moderate Alzheimer’s disease. An 18-month observation demonstrated that Lecanemab reduced cognitive dysfunction by 27% on the primary endpoint. Finally, the drug received full FDA approval [23,24]. Due to the links between AD and aging, the urgent necessity for novel therapeutic drugs targeting cellular senescence has become increasingly apparent. Preclinical studies with senolytic drugs, such as Dasatinib and Quercetin (D + Q), have indicated positive findings, including improvements in cognitive decline, reduction in amyloid and pathological tau accumulation, as well as decreased neuroinflammation in mouse models [25].

## 3. Cellular Senescence and Alzheimer’s Disease

### 3.1. General Information

Aging is a process marked by time-dependent progressive deprivation of tissue as well as organ function, which results in increased risk of developing numerous human pathologies, including neurodegenerative disorders. Recently, great emphasis is being placed on cellular senescence mechanisms as one of the major factors in a heterogeneous aging process [6]. Cellular senescence (CS) is defined by an irreversible arrest of proliferation in dividing cells (postmitotic cells also show senescence without any arrest) and can be caused by a number of triggers [26]. Cell cycle arrest is controlled by the p16/Rb and the p53/p21 pathways [27]. As a result of an increased level of p53, upregulation of p21 occurs and arrests the cell cycle. The p16INK4a activation leads to inhibition of CDK6 and cyclin-dependent kinase 4 (CDK4) that hypophosphorylates pRB in order to stop S phase entry. This leads to irreversible cell cycle arrest [27]. Mobilization of oncogenes, for example, *HRAS* and *BRAF*, leads to DNA damage, and as a result normal cells become senescent [28]. We can distinguish stress-induced senescence, dysfunctional mitochondria-induced senescence, telomere-dependent replicative senescence, oncogene-induced senescence, oncogene overexpression, for example, Ras, E2F1, Cdc6, Mos, Akt, and cyclin E, and DNA damage-induced senescence [29]. Senescence-associated secretory phenotype (SASP) is a secretion of pro-inflammatory molecules, which exhibit paracrine effects on neighboring cells (this can lead to alteration in cellular metabolism and cause toxicity). Examples of secreted molecules include chemokines, cytokines, microRNAs, bioactive lipids, extracellular matrix proteases, and extracellular vesicles [30]. SASP is associated with many processes, such as exacerbated inflammation; it was also found to promote angiogenesis, favor cell proliferation, and ignite the fibrogenic cascade [26]. Cyclic GMP–AMP (cGAMP) synthase (cGAS)—stimulator of interferon genes (STING) pathway might have a leading role among many signaling pathways in the SASP [28]. Senescent cells accumulate in tissues with age. Combined with the detrimental effect of SASP, they might lead to significant damage, elevated risk of chronic diseases and mortality [30,31]. The morphology of the senescent cells differs from normal cells. These cells no longer have their typical features, and their size expands mainly as a result of cell cytoskeletal rearrangements [29,30]. Atypical appearance is also visible in the nucleus, mitochondria, and endoplasmic reticulum (ER) [30]. Currently, there is no universal marker that identifies senescent cells and distinguishes them from other non-proliferating cells. In an attempt to identify senescent cells, combinations of biomarkers are used, such as markers of arrest of cell cycle (upregulation of p16 and p21 and decreased levels of the proliferative marker Ki67, DNA synthesis and lamin B1 expression), triggering of DNA damage response signaling (formation and accumulation of γH2AX foci), and increased lysosomal activity (detection of lipofuscin or SA-β-gal). RNA and protein levels of SASP factors (including interleukins IL-1α/β, IL-6, and IL-8, matrix metalloproteinase MMP1, cytokine GDF15, and chemokine CXCL1) are also analyzed [32]. The senescence-associated-β-galactosidase (SA-β-gal) staining procedure is known to be the gold standard test for confirming senescence [29]. Elevated SA-β-Gal action in nervous system cells (for example, senescent astrocytes) has been found in in vitro as well as in vivo studies [33]. Senescent cells remain metabolically active and exhibit excessive secretion. They trigger inflammation. SASP attracts immune cells and enables them to reach the impaired area. Senescent cells have been shown to be deeply associated with chronic inflammatory disorders [34], aging, and disorders related to age, for example, musculoskeletal dysfunctions (including osteoporosis, osteoarthritis and sarcopenia), cancer, diabetes, and fibrosis [32]. SASP is a key contributor to the onset and progression of neurodegenerative diseases. Interleukin-1β (IL-1β) is a pro-inflammatory cytokine that is part of the SASP and can be induced by Aβ and tau oligomers (TauO) in the brain [35]. Senescent cells are disease-specific and can be very heterogeneous. Later in life, when senescent cells accumulate due to many mechanisms, senescence is suggested to have harmful effects [36]. There are studies indicating a link between cellular senescence and the development of AD. Various studies have revealed molecular pathways which show a connection between the atypical accumulation of beta-amyloid and hyperphosphorylated tau proteins and dysfunction of mitochondria, oxidative stress, genomic instability, and abnormal synaptic transmission [37]. Cellular senescence, demonstrated through DNA damage, higher level of SA-β-gal, p53 level, elevated release of components of the SASP, and telomere damage, has been observed in brain cells of patients with AD. Examples of cell types include astrocytes, neurons, and microglia [27]. Senescent cells exhibit permanent cell cycle arrest and are resistant to apoptosis, and changes by the DNA damage response (DDR) also occur. Apoptosis avoidance occurs through upregulation of DDR-related proteins, such as γH2AX. One major source of inflammation that plays a role in the neurodegenerative process, is the SASP, a secretory phenotype producing inflammatory proteins and matrix-degrading enzymes. SASP components induce apoptosis (although they have the ability to inhibit apoptosis in specific circumstances). In addition, senescent cells secrete SASP factors which render neighboring cells that are also senescent in a paracrine fashion. This process is called the bystander effect [38]. Recently the role of tau and Aβ aggregates in triggering senescence in Alzheimer’s disease has been noted (predominantly by activating DNA damage response signaling) [35]. On the other hand, it has been shown that tau oligomers (TauO)-induced SASP components enhance neuroinflammation, tau pathology, and neurodegeneration [35]. This occurs through, for example, interleukin-1β (IL-1β), a component of SASP that can be induced by Aβ and tau. IL-1β activates microglia and astrocytes to further produce inflammatory mediators, potentially leading to synaptic dysfunctions and, consequently, memory disturbances. Furthermore, IL-1β affects the expression of *N*-methyl-D-aspartic acid (NMDA) receptors, which are crucial in cognitive processes, including learning [35]. The causal connection between Alzheimer’s disease and cellular senescence remains unclear. It is suggested that oxidative stress and dysfunction of mitochondria lead to DNA strand breaks that trigger cellular senescence. Senescent cells take part in the formation of beta-amyloid plaques, which creates a vicious circle cycle [16]. Studies show that exposure to the toxic formations of Aβ or aggregated tau is sufficient to induce senescence in various types of brain cells, whereas selective eradication of senescent cells reduces the accumulation of Aβ and the formation of neurofibrillary tangles (NFTs) in mouse brains, improving learning and memory [7]. In studies on mouse models of tau-dependent neurodegenerative disorders, accumulation of senescent astrocytes and microglia has been noted [39]. Research was conducted on mouse models P301S, P301L, hTau, and 3xTg-AD, which served as models of amyloidopathy or tauopathy in Alzheimer’s disease, along with control mice (age-matched) for each strain. The authors of the study measured the levels of cellular senescence markers in the brains of the mice and demonstrated that these markers were elevated in the hTau and P301S models at the onset of Alzheimer’s-like disease, suggesting that the increase in senescent cells preceded the onset of the disorder in these mice [40].

### 3.2. Alzheimer’s Disease and Senescence of Astrocytes

An example of nervous system cells impacted by cellular senescence are astrocytes. In the case of astrocytes, cellular senescence markers consist of growth arrest, increased SA-β-Gal activity, and increased expression of senescence-associated genes *p53* and *p21^WAF]^* [33]. Worth mentioning is the fact that astrocytes have been found to be more vulnerable to senescence-inducing stimuli than several other cell types, for example, fibroblasts. Pathologies of the astrocytes lead to the initiation and progression of AD mostly because of astrocytic neuroinflammation. Recent studies show a significant impact of astrocyte senescence in the development of Alzheimer’s disease [33]. In animal model studies of tau-dependent disorders, accumulation of both senescent microglia and astrocytes was found in cortices and hippocampi [41]. Inflammatory and proteolytic SASP factor levels secreted by senescent astrocytes include IL-6 (main SASP component and a biomarker of AD), IL-1β, CCL2, TNF-α, MMP-3, MMP-9, MMP-10. These SASP are elevated in the cerebrospinal fluid and sera of AD patients [33,41].

Impact of senescent astrocytes on brain functioning in AD has been summarized in Table 1.

### 3.3. Neurons, Cellular Senescence, and Brain Inflammation

Post-mitotic cells such as neurons can undergo a senescence-like state, contributing to brain aging and neurodegenerative diseases [5,57]. This process also affects other types of cells, such as endothelial cells or even fibroblasts. Senescent endothelial cells are enlarged and are resistant to phenotypic changes [58]. Interestingly, accumulation of Aβ also occurs in the skin, particularly in fibroblasts. Skin fibroblasts derived from patients with AD display multiple functional abnormalities, including aberrant activation of bradykinin receptor signaling, dysregulated cholesterol ester metabolism, disturbances in calcium homeostasis, and compromised mitochondrial activity [59]. Neurons, despite being non-dividing, exhibit many classical hallmarks of senescence, including DNA damage, mitochondrial dysfunction, altered metabolism, high ROS production, increased p16 and p21CIP1 levels, activation of the p38MAPK pathway, and the secretion of pro-inflammatory factors, collectively known as the senescence-associated secretory phenotype (SASP). These dysfunctional neurons accumulate with age and may promote chronic neuroinflammation and neuronal network disruption, thereby facilitating the onset and progression of neurodegenerative disorders such as Alzheimer’s disease [9,31,57]. The post-mitotic neurons, that undergo senescence (amitosenescence), express markers such as SA-β-gal, MCP-1, γ-H2AX, 4-HNE, and exhibit factors of senescence faster than glial cells [9]. The research conducted by Dehkordi et al. using the 5 × FAD mouse model provided insights into the cellular identity of senescent cells in the hippocampus. Immunohistochemical co-localization of p16 with cell-specific markers revealed that p16 was predominantly observed in neurons. Importantly, neuronal senescence was evident even at early stages of AD, suggesting its potential role as an early indicator of AD etiopathology [60]. Senescence in non-dividing cells involves a persistent DNA damage response (DDR) signaling, particularly at telomeric regions, even in the absence of cell division. This form of “senescence without proliferation” may explain how aging neurons display stress markers and contribute to tissue dysfunction despite not dividing [9]. Furthermore, senescent neurons have emerged as a potential neurogenic source of chronic brain inflammation observed in late stages of AD. Even a small population of senescent neurons may exert disproportionately large effects on brain function and inflammatory responses. A single neuron forms thousands of synaptic connections, thereby facilitating the widespread propagation of dysfunction through SASP-related mechanisms [61]. Findings demonstrate that senescent neuronal cultures derived from AD models acquire a pro-inflammatory SASP capable of inducing reactive astrogliosis (abnormal increase in the number of astrocytes and their hyperreactivity) [61]. To evaluate astrocyte reactivity in response to SASP factors secreted by AD-induced neurons, Herdy et al. performed RNA sequencing on human astrocyte cultures treated with conditioned media derived from control neurons and AD-induced neurons. The researchers employed a non-contact co-culture system. They collected conditioned media (CM) from six AD and six control iN (induced neurons) cultures and applied it to healthy primary human cortical astrocytes (ApoE3 homozygous). This setup did not involve direct neuron–astrocyte co-culture in order to isolate the effect of neuron-secreted factors on astrocytes [61]. Genes typically associated with reactive astrogliosis were significantly upregulated, such as *nestin* and *vimentin* involved in cytoskeletal remodeling; *ALDOC*, *FABP7* participating in metabolic regulation; *CRYAB* marked in chaperone activity; *C3*, *Serpina3n*, *IL6*, the pro-inflammatory mediators; and signaling receptors such as *NTRK2*, *IL17R*. A marked downregulation of glutamate transporter genes (*EAAT1*, *EAAT2*), and *TIMP3* were observed. The transcriptional alterations observed in these astrocytes closely parallel those reported in astrocytes from post-mortem AD brain tissue [61].

### 3.4. The Role of Senescent Microglia

Microglial cells are considered to be the resident macrophages of the central nervous system (CNS), and play a key role in regulating neural homeostasis [16]. As part of the innate immune system, they regulate the number of neurons in the CNS, and participate in synaptic formation associated with learning [16]. To maintain homeostasis, microglia are in continuous interaction with neurons. These interactions occur via direct elements of the synapse, through microglia–neuron communication mediated by pannexin-based channels and connexin-based gap junctions, or by releasing exosomes and microvesicles [62]. Microglia also interact with astrocytes which plays a role in homeostasis, and possibly neurodegeneration [63]. Activated microglia release factors such as IL-1α, IL-1β, TNF-α, and C1q, which in turn induce astrocytes to transition into a toxic (A1-like) reactive phenotype. This transformation reflects an astrocytic response to microglial cues in a way that exacerbates pathology rather than supports repair [64]. Microglia undergo apoptosis which can be activated extrinsically or intrinsically. In the extrinsic pathway, death receptors are activated through the binding of extracellular ligands. In the intrinsic pathway, stimuli, such as DNA damage, activate p53 resulting in the upregulation of pro-apoptotic factors in the Bcl-2 family [65]. Senescent microglia exhibit altered functionality and SASP expression, and they are also resistant to apoptosis [66]. In studies on the pathology of Alzheimer’s disease development, microglial phenotypes are differentiated [16]. Phenotypic features may be categorized into the resting phase M0 and the activated phases M1 and M2 [16]. The M1 phenotype, known as pro-inflammatory, is classically activated in response to lipopolysaccharide (LPS) or interferon-γ (INF-γ), initiating the release of neurotoxic factors associated with the inflammatory response. The anti-inflammatory M2 phenotype, by comparison, is activated by IL-4 or IL-13 stimulation and secretes neurotrophic factors such as IGF-1, VEGF, and anti-inflammatory cytokines, that promote neuroprotection [67]. Upon activation by ligands such as infectious pathogens, amyloid-β, or aggregated α-synuclein, NADPH produces superoxide, which, through further transformations, leads to necrosis or apoptosis of neurons [63]. Cathepsins, are released from lysosomes in response to Aβ, resulting in neuronal apoptosis [63]. There are immunological pathways that prevent excessive activation in response to external stimuli; however, their dysregulation and disturbances at control points initiate or exacerbate neurodegeneration [63]. Aging is believed to significantly influence the proportion between M1 and M2 phenotypes. The aging process enhances the M1 microglial phenotype, characterized by elevated levels of pro-inflammatory cytokines such as IL-1β and TNF-α, while reducing M2 microglial activation. Amyloid-β-induced M1 microglia are regulated by the receptor for advanced glycation end products (RAGEs) and Toll-like receptors (TLRs), while pro-inflammatory cytokines further activated astrocytes, promoting neuronal loss in AD [67]. Growing evidence indicates that aging triggers a morphological shift in microglia from a branched state to a spherically activated phenotype, exhibiting abnormal cytoplasmic structure. A hypersensitive phenotype is observed, showing an excessive response to neurotoxic and inflammatory factors [68]. The role of microglia in the pathogenesis of Alzheimer’s disease is also supported by genetic aspects, such as certain rare variants of TREM2 (triggering receptor expressed on myeloid cells-2), which significantly raise the risk of developing AD. Studies have observed increased expression of TREM2 in AD patients, which appears to be linked to the migration of microglia toward amyloid plaques [69]. Furthermore, aging contributes to the upregulation of TREM2 [69]. Amyloid β modulates numerous microglial responses, including activation and proliferation, and these responses contribute to the process of limiting further amyloid deposition. The interaction between Aβ and several microglial receptors, including TREM2, is what mediates Aβ phagocytosis. Microglia detect and engulf Aβ aggregates through surface receptors like scavenger receptors (SRs) and Toll-like receptors (TLRs). For instance, TLR2 activation enhances the uptake of amyloid β peptides [68]. Aging of microglia may sensitize them to inflammatory indicators and exacerbate the Aβ pathology underlying Alzheimer’s disease. Furthermore, senescent microglia have been shown to contain hyperphosphorylated tau protein. It is suggested that aging impairs the clearance mechanisms, leading to intracellular accumulation of pathological tau proteins [68].

### 3.5. Alzheimer’s Disease and Dysregulation of Cellular Autophagy

There are several principal mechanisms for removing damaged proteins. Proteasomes are macromolecular assemblies comprising a 20S core particle, a 19S regulatory particle, and alternative regulatory caps, which together degrade specific classes of misfolded proteins. Protein degradation can proceed via a targeted, ubiquitin-dependent pathway. Ubiquitin is a 76–amino acid protein that can be conjugated to other proteins as a post-translational modification in a process called ubiquitination. In addition, a ubiquitin-independent mode of degradation occurs via proteolysis by the free 20S proteasome, whose primary substrates include oxidatively damaged proteins generated under oxidative stress. Another equally important mechanism for removing impaired proteins is autophagy [70]. Protein aggregates and damaged organelles are engulfed into autophagosomal membranes forming vesicles called autophagosomes that then lyse with lysosomes and their membranes. The pathway of this autodegradation can be activated as a result of tissue damage and biological stress. In animal models, autophagy exhibits neuroprotective functions in relation to microglia. Conversely, the deterioration of microglial autophagy aggravates tau accumulation in a tauopathy model [71]. This process has been proved to be impaired in individuals with neurodegenerative disorders, including AD. The formation of autophagic vacuoles (AVs) in cells follows a defined sequence, starting with the formation of autophagosomes, progressing through amphisomes to autolysosomes [72]. In Alzheimer’s disease, however, all forms of AV accumulate in affected neurons. The extent of autophagic vacuole accumulation resembles some primary lysosomal disorders (LSDs). MLBs, which are multilamellar bodies, are indicators of disrupted lipid and cholesterol metabolism. They were found in dystrophic neurites in patients with AD, they also concentrate inside cells in pathological conditions as lysosomal disorders [73]. Autophagy is triggered by several factors, including decreased insulin and ATP levels [74]. The cell’s energy status, expressed as the AMP/ATP ratio, is detected by the mammalian target of rapamycin complex 1 (mTORC1) and adenosine monophosphate-activated protein kinase (AMPK) [74]. Signal downstream to the serine/threonine-protein kinase ULK1, (unc-51-like autophagy-activating kinase 1) complex to assemble the class III phosphatidylinositol 3-kinase (PI3K) complex I to the phagophore assembly site, where VPS34 kinase, along with Beclin-1, VPS15, and ATG14 creates phosphatidylinositol 3-phosphate (PI3P) [74]. A key molecular action in the process of autophagy is the fusion of LC3 family members with phosphatidylethanolamine on the phagophore membranes [74]. Impaired autophagy can contribute to aging as a result of defective p62-dependent selective autophagy of the transcription factor GATA4. GATA4 levels and markers of cellular senescence increase in aging human brains and show spatial correlation in oligodendrocytes, astrocytes, and pyramidal neurons [74]. Tau protein and amyloid-β are substrates of autophagy, and their levels are directly controlled by it. In the brains of individuals diagnosed with AD, various alterations can be observed, such as hyperactivation of the mTORC1 complex, which inhibits autophagy, and suppression of the downstream ULK1 complex that promotes autophagy. Furthermore, Beclin 1, a key component of the autophagy-initiating PI3K complex I, is markedly reduced in the brains of individuals with AD. The activity of serine/threonine-protein kinases like PTEN-induced kinase 1 (PINK1), along with ULK1, and TANK-binding kinase 1 (TBK1) is also decreased [74]. A summary of autophagy and the impact on Alzheimer’s disease is presented in Figure 1.

### 3.6. Defective Mitochondria

An association between mitochondrial dysfunction and Alzheimer’s disease has been demonstrated. The increased presence of impaired mitochondria is a hallmark of aging and age-related neurodegeneration [75]. Mitophagy, a specific selective type of autophagy where the substrate are defective mitochondria, is a cytoprotective mechanism responsible for removing unnecessary or dysfunctional mitochondria to maintain intracellular homeostasis [76]. Mitochondria are cellular organelles responsible for ATP production. However, by-products such as reactive oxygen species (ROSs) and hydroxyl radicals are also produced during processes related to cellular metabolism, leading to oxidative stress. The high energy demands of nerve cells make neurons particularly dependent on mitochondrial processes. Therefore, impairment of their appropriate function contributes to the degeneration of nerve cells. The characteristic amyloid -β and abnormal tau proteins associated with AD interact directly with mitochondria. Increased expression of *Drp1* and *Fis1*—mitochondrial structural genes involved in their fission—has been observed, while Mfn1, Mfn2, and OPA1, which are fusion proteins, show reduced mRNA levels. Together, these changes lead to increased fission and fragmentation of these organelles. The mitophagy process is disrupted and unable to keep up with the demand for the clearance of defective mitochondria [77]. Under normal conditions, PINK1 is degraded. Upon mitochondrial depolarization, PINK1 accumulates on the outer mitochondrial membrane (OMM) and recruits Parkin, initiating mitophagy. In AD, this system is dysfunctional. Parkin becomes depleted from the cytosol as the disease progresses, leading to impaired mitophagy despite signs of its activation [78]. Several proteins crucial for the proper progression of mitophagy and selective kinds of autophagy have been described, including PINK1, Parkin, ULK1, and TBK1. Reduced mitophagic flux is related to defects in the initiation of the PINK1/Parkin cascade. Increased levels of PINK1 in the primary stages of Alzheimer’s disease (AD) and elevated Parkin levels in the final stages of the disease have been detected in the hippocampi of patients with sporadic AD [79]. Additionally, reduced levels of proteins associated with mitophagy and autophagy, such as optineurin, Beclin-1, ATG5, ATG12, BNIP3, and FUNDC1, have been observed [79]. Increased mitophagy is beneficial and contributes to delaying disease progression and enhancing survival, as demonstrated in studies on nematode models [80]. Various mitophagy inducers, such as NAD+ precursors, spermidine, and urolithin A, have been proven to have substantial benefits in augmenting mitochondrial resistance against oxidative stress and enhancing mitophagy in both animal models and human cells which shows that the stimulation of mitophagy and might have a crucial role in Alzheimer’s disease therapy [78].

### 3.7. Therapeutic Options Targeting Cellular Senescence

As previously described, senescent cells secrete various pathogenic SASP factors which can induce senescence in neighboring cells. They are also resilient to cell death. This has led to a strong focus on developing therapies aimed at eliminating senescent cells [30]. The term senotherapies refers to numerous approaches targeting senescent cells [81]. Promising results in delaying dysfunction and extending healthspan have been shown by recent research [82]. One form of intervention is lifestyle change. Exercise has been shown to prevent obesity-induced senescent cell formation in animal model studies [30]. Dietary interventions can take many forms, such as calorie restriction and intermittent fasting. Caloric restriction has been shown to prevent the formation of senescent cells. Possible mechanisms by which this effect is achieved involve autophagy activation, reduction in ROS, and triggering of DNA damage repair [30]. Another form of intervention is pharmacotherapy. First-generation senotherapeutics (senolytics) target numerous signaling pathways, for example, growth factor receptors, tyrosine kinase receptors, molecules like BCL-2, p53 modulation, and inhibition of caspases. This group includes Dasatinib, Quercetin, and fisetin. Second-generation senotherapeutics consist of senomorphics (SASP inhibitors), SA-β-gal-activated prodrugs, lysosomal activators, CAR-T cells immune-mediated clearance, and many more. This group includes ruxolitinib, metformin, and olokizumab [82]. Senolytics trigger apoptosis which leads to removal of the senescent cells. At the moment there are over 20 senolytics that have the ability to selectively destroy senescent cells in mice and humans, for example, Dasatinib, fisetin, and Quercetin [30,83,84]. In recent mouse model studies, aged mice with an increased number of p16Ink4a-positive senescent cells have been treated with a senolytic cocktail of Quercetin and Dasatinib. As a result, a decrease in p16Ink4a exclusively in the microglial population has been noted. This led to reduced activation of microglia and SASP factors expression. Cognitive function also improved under this treatment [85]. In other animal model studies, it has been suggested that elimination of senescent microglia and astrocytes by genetic ablation or senolytic agents could prevent or inhibit neurofibrillary tangles formation and neurodegeneration in Alzheimer’s disease and tauopathy [39,56]. Worth mentioning is rapamycin, an inhibitor for the mammalian target of rapamycin (mTOR). Rapamycin is a serine/threonine kinase that controls the eukaryotic cells response to growth factors, nutrients, and cellular energy status [86]. Recently it has been demonstrated that mTOR, as a part of the PI3K-AKT-mTOR-ROS pathway, is obligatory for CKII (protein kinase CK2) inhibition-mediated cellular senescence. Through decreased creation of reactive oxygen species (ROSs), rapamycin blocks CKII downregulation-mediated senescence [87].

## 4. Conclusions

Senescence of organisms is inevitable, and aging not only occurs in organs and tissues but also at the cellular level, affecting even the smallest structures, such as organelles. Delving into this issue, the link between aging and the development of neurodegenerative diseases—with Alzheimer’s disease at the forefront—stands out. Studies on human and animal cells have revealed that senescent cells are characterized by a modification of metabolic functions, resistance to apoptosis, and abnormal morphology. This also applies to cells of the central nervous system: neurons, microglia, and astrocytes, in which senescent cells have been found in the brains of Alzheimer’s disease patients and animals. Identification of senescent cells is provided by numerous biomarkers, such as markers of cell cycle arrest, activation of DNA damage response signaling, and increased activity of organelles involved in cellular recycling—lysosomes. Although cellular senescence has positive aspects, such as involvement in tissue remodeling or wound healing, it also plays a role in inflammation, contributing to chronic inflammatory diseases, cancer, and neurodegeneration. The link to neurodegeneration has been demonstrated through studies on beta-amyloid. Both the accumulation of Aβ in the course of the disease triggers cellular senescence, and senescent cells contribute to its deposition. Furthermore, aging-related processes involving mitochondria and cellular organelles, as well as dysfunctions in autophagy, play a role in Alzheimer’s disease. Cellular senescence also disrupts the balance between different microglial phenotypes, promoting a pro-inflammatory type that contributes to neurodegeneration. Considering the changes occurring due to the aging of both the organism and its cells, it is clear that Alzheimer’s disease requires exploration on multiple levels.

## Figures and Tables

**Figure 1 ijms-26-08638-f001:**
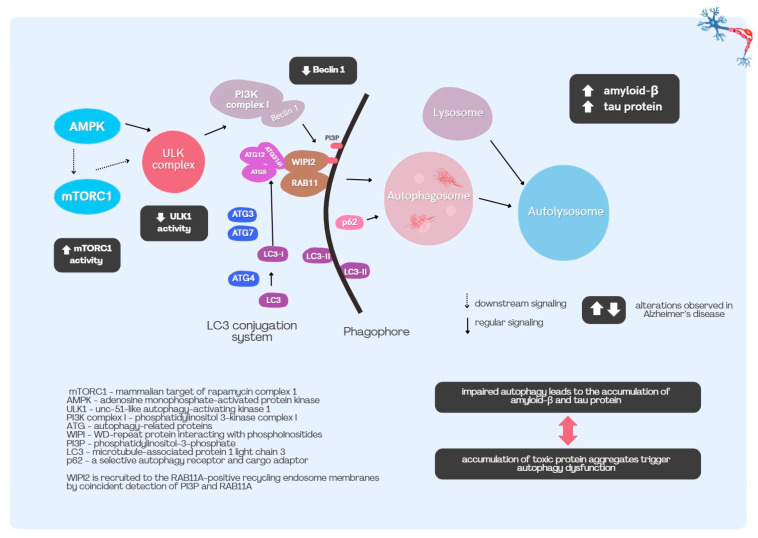
Autophagy and the impact on Alzheimer’s disease on its course.

**Table 1 ijms-26-08638-t001:** Functional consequences of senescent astrocytes in AD.

Senescence Associated Changes	Functional Consequences
Accumulation of Aβ	Expression of APP and Aβ generation in neurons can be increased by senescent astrocytes secreting SASP factors [42]. Reduction in expressions of LRP1 and SR-B1 which suggests that senescent astrocytes show reduced ability to uptake and degrade Aβ [43,44]. Increased secretion of SASP by astrocytes can reduce glia ability to stimulate clearance of Aβ and facilitates its accumulation in the brain [33].
Promotion of Chronic Inflammation and Activation of Microglia	SASP factors secreted by senescent astrocytes prone to induce inflammation include, for example, IL-6, IFNγ, TGFβ, and CXCL10 [45,46]. IFNγ is a potent regulatory cytokine with the ability to activate microglia and promote inflammation in the brain of patients with Alzheimer’s Disease [47,48].
Dysfunction of Blood–Brain Barrier	Various SASP factors produced by senescent astrocytes influence the permeability of the blood–brain barrier. Examples of SASP influencing the BBB include IL-6, transforming growth factor-β (TGFβ), basic fibroblast growth factor (bFGF), and glial cell–derived neurotrophic factor (GDNF) [49].
Neuronal Loss and Synaptic Dysfunction	In the CNS of patients with Alzheimer’s disease senescence-like phenotype of astrocytes is shown around Aβ plaques and NFTs [45,50,51]. Dysfunction in synapses is also predominantly found surrounding dense-core Aβ plaques [52]. Connection between senescent astrocytes and synaptic dysfunction or synapse loss is notable in advancement of AD [53]. Release of SASP factors, for example, IL-6, can prompt the neuronal loss which is observed in patients with AD [44]. Decreased secretion of neurotrophins (for example: nerve growth factor (NGF), and brain-derived neurotrophic factor (BDNF), by senescent astrocytes [54]. Contribution of senescent cortical astrocytes in the cognitive decline and impairment of synaptic plasticity because of a decreased production of ATP (generated mainly in mitochondria) [55].
Formation of NFT and Accumulation of Tau	Various studies have shown the impact of senescent astrocytes in the formation of NFT and hyperphosphorylation of tau [39,56]. Recent animal model studies have shown the pivotal role of senescent astrocytes in the accumulation of tau. They also demonstrated a strong connection between astrocyte senescence, formation of Aβ, and accumulation of tau. Unfortunately, the underlying mechanism is unclear and inconclusive [39].

## Data Availability

Data sharing is not applicable, as no datasets were generated or analyzed during the current study.

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
