# Peer review of "Connections Between Cellular Senescence and Alzheimer’s Disease—A Narrative Review"

_ijms, 2025, doi:10.3390/ijms26178638_

Round 1
Reviewer 1 Report
Comments and Suggestions for Authors
The narrative review titled "Connections between Cellular Senescence and Alzheimer’s Disease" by Kuzniar et al. aims to highlight the links between cellular senescence and Alzheimer’s disease (AD). The review begins by outlining that AD is the most common form of dementia, with ageing as its primary risk factor. At the cellular level, ageing involves the progressive loss of function, and one hallmark of this process is cellular senescence, a state in which cells permanently exit the cell cycle after a finite number of divisions.
Cellular senescence can be triggered by various stressors, including DNA damage, telomere shortening, mitochondrial dysfunction, oxidative stress, and oncogene activation. In addition to growth arrest, senescent cells are characterized by the secretion of a variety of pro-inflammatory cytokines/molecules, collectively termed the senescence-associated secretory phenotype (SASP). The composition of the SASP can vary depending on the cell type and context.
In their review, the authors assert that AD is a neurodegenerative disease with links to cellular senescence, citing evidence that senescent cells accumulate near amyloid-beta (Aβ) plaques and neurofibrillary tangles. They discuss several mechanisms through which senescence might contribute to AD pathology, including astrocyte senescence, mitochondrial dysfunction, impaired autophagy, and activation of senescent microglia.
However, the overall message of the review is unclear. While the authors provide an overview of relevant literature, the narrative lacks a clear, synthesised argument about how senescence contributes to AD. Are they suggesting that cellular senescence plays a causal, contributory, or reactive role in disease progression? I would agree that senescence is involved in AD, but it’s not clear from the review what is the central mechanism? Is the key driver the secretion of SASP factors, particularly pro-inflammatory cytokines, that exacerbate neuroinflammation?
A critical point the review could address more clearly is: which cell types undergo senescence in AD? Evidence suggests that multiple CNS cell types may be involved, astrocytes, microglia, endothelial cells and potentially even neurons and fibroblasts. All of them could contribute SASP components, but the composition and impact of the SASP likely varies by cell type – this is not really addressed in the review.
Another important question is whether ageing and senescence are causative in AD. Ageing is certainly a major risk factor, but it cannot be the only cause, otherwise all elderly individuals would develop AD. How do we reconcile this with cases of early-onset Alzheimer’s disease, where ageing per se is not the primary driver?
In my view, the strongest mechanistic link between cellular senescence and AD is likely the secretion of SASP factors, which can intensify neuroinflammation. However, it remains unclear whether SASP secretion is a cause of pathology or a response to existing damage, such as Aβ plaques and tau tangles. It’s plausible that these pathological hallmarks act as stressors that induce senescence in neighbouring cells, creating a feed-forward loop of SASP secretion and inflammation.
Overall, while the review successfully compiles existing literature, it does not fully synthesize the data into a cohesive model. A more structured argument would benefit readers, especially those in neuroscience who may be less familiar with senescence biology. The ageing research community is well aware of the implications of the SASP, but this concept may not yet be widely appreciated in the context of neurodegeneration. A clearer articulation of how senescent cells drive brain inflammation via SASP and which cells are responsible could make this an invaluable contribution to the field.
Reviewer 2 Report
Comments and Suggestions for Authors
This is a generally interesting review with some merit but it has a large number of scientific shortcomings and thus requires a number of corrections including language. Importantly, the authors have to clearly discriminate between inherited AD cases with known mutations and the vast majority of spontaneous AD. In addition, the authors need to mention that also postmitotic (PM) cells such as neurons show senescence but without any cell cycle arrest. This is just one of various examples where senescence-related processes are incorrectly described, including to mix cause and consequences sometimes. Moreover, the entire review contains just 1 table which would be better presented as a normal text and 1 figure which seems rather a low number. Importantly, the table content requires references and the figure just shows a completely general process of autophagy without any specific relation to AD although it is claimed in the figure title. A number of other statements also lack required references or just cite other general reviews instead of specific original papers/studies. Importantly, while different brain cells play a role in AD, the main cell type-neurons-have not been described at all under the topic of senescence by the authors although there are thousands of papers on that topic which is a rather serious shortcoming of the review which is rather superficial both in the fields of senescence as well as AD and brain-related cell types. This superficiality is also documented by the fact that the authors mainly cite other reviews instead of original studies/references. The structure of the review seems rather erratic, jumping from various cell types to subcellular organelles, processes etc. and then back to other cell types. This needs to be improved for a more logical flow. Please see more details below:
- In my view, instead of “cellular aging” it is better to say “cellular senescence” throughout the ms since, as you correctly defined, aging rather refers to whole organisms while on the cellular level the correct term is “senescence”,
- Line 18: Please remove “the”.
- Line 22: There is mainly 1 SASP which can consist of different factors, so I suggest to use it in singular only.
- Line 23: please add “a” in front of “neurodegenerative”.
- Line 24: The term “proven” is not a good term, better use “shown/demonstrated”.
- Ine 33: Please remove “The” in front of “research.
- Line 37: correct grammar is “consisting”.
- Line 40: Please remove “the” in front of “changes”.
- Line 41: : correct grammar is: “symptom onset” or “onset of symptoms”.
- Line 44: “points TO...”
- Line 45: decline of WHAT? Please specify. Also, please replace “the” with “a” in front of “reduced”.
- Line 52: Please clarify what you mean with “regeneration ability” which is applicable to tissues, but rather not to cells.
- Lines 53: as mentioned before “SASP” in singular.
- Line 54: a phenotype cannot be released, it should be “SASP factors” when you refer to physical entities while “phenotype” is something abstract.
- Line 56: Please provide a reference for the statement. Also, it should be “MOUSE brain”, in complex nouns, the first should be always in singular.
- Line 57: Please provide a reference for the statement and amend “cell aging”, see comment above.
- Line 61: Please remove “tried” since it sounds as if it was not successful and say “We summariseD”.
- Line 62: cellular senescence cannot be an approach. Please rephrase saying, for example ”TARGETING senescence...”.
- Line 66: You wrote a review, which is not research. Please replace “research” with “SEARCH”.
- Line 68: What do you mean with “short reminder”? OF WHAT? Please clarify.
- Line 69: Why “national” concern? It is rather a world-wide one.
- Line 71: References 10 and 11 are not about epidemiology. Importantly, please specify where the 152 million patients occurred-in a specific country (ref. 9 is about China), a region or world-wide?
- Lines 72/73: AD is not the same as dementia, so please specify where the costs apply to- in a specific country, a region or world-wide? Again, reference 10 is not about epidemiology and you should not cite data from an introduction of other papers without giving the primary sources.
- Line 77: ApoE4 is a variant in itself and to my knowledge does not require a mutation.
- Line 84: Please provide a reference.
- Line 85 and other places: Please emphasise that only a minority of AD cases has a genetic component with mutations. The vast majority is sporadic without any known mutations, but other risk factors such as specific APOE types etc.
- Line 87: Please specify “signal transduction”. Do you refer to synapses?
- Lines 84/85: As mentioned above, mutations only refer to a small group of inherited AD cases. Please amend throughout the ms.
- Line 93: Please remove “the” in front of “pro-“ and add a dash there.
- Lines 99 and 101: add a dot at the end of the sentences.
- Line 100: Please replace “receive” with “SHOW” since the former is not correct language.
- Line 113: I would not call it “focus” since the reason is the lack of a cure due to still unknown underlying causes of AD.
- Line 114: Please add “mainly” after “Therapy”.
- Line 131: Please add “hyperphosphorylated” or “pathological” in front of tau since normal tau does not accumulate.
- Line 139: Proliferation arrest occurs only in dividing cells while postmitotic (PM) cells also show senescence without any arrest. This is important, for example for PM neurons.
- Line 141: You incorrectly describe the pathways: IT is the p16/Rb and the p53/p21 pathways. Please correct this scientific mistake and also provide a reference. Moreover, it is not a GENE expression level that regulates p53 activity, but its conformation of the tetra-structure. Importantly, use the term “expression” only for genes and use “level” for RNA and proteins.
- Lines 150 and 152: SASP is a secretin pathway and not related to exocytosis which is differently defined and involves vesicles. Please correct throughout the ms.
- Line 157: Please replace “character” with “role”.
- Lines 161/2: It seems that “deflation” and size increase contradict each other. Please remove the former which was never described in senescent cells and thus seems an unsuitable, unscientific term.
- Line 162 “membrane change”, see explanation above.
- Line 163: Please add “the” in front of “nucleus”.
- Line 168: 53PB3 is not regulated on the level of gene expression and the protein is there always, it just forms foci after DNA damage. Please correct
- Line 170: Firstly, do not use “expression” when referring to proteins. Secondly, you probably mean FACTORS, NOT fragments. Please correct.
- Line 177: Please specify what you mean with “problem area” which is not a scientific term.
- Line 179: It is not clear why you give PD here as an example of neurodegenerative diseases (NDDs) when your review is about AD.
- Lines 182, 192 and others: There are no “aging cells”, just “senescent cells”. Please correct.
- Line 184: It is not correct to call senescence a “hallmark of carcinogenesis” since, in addition to be a tumour SUPPRESSOR, it is a secondary response to cancer treatment as reference 34 clearly states and not a hallmark. Please correct.
- Lines 185, 204, 205, 212, 215 and many others: Please replace “aging” with “senescence”. See explanation above.
- Line 189: Please replace “bigger” with “higher” since language is incorrect. Moreover, both sen-beta-gal and p53 are not genes, thus, the use of “expression” is incorrect here.
- Line 190: Please add “the” in front of SASP.
- Line 193: Your statement sounds as if senescent cells are resistant to DDR which you probably do not mean. Please rephrase and correct language structure.
- Lines 196/7: Please provide a reference and it is not correct that the only inflammation stems from the SASP, there are other inflammatory processes in AD brains that also contribute to neuroinflammation, for example microglia-dependent inflammation and many other processes. Please correct the false statement. Importantly, in the next sentence you describe inflammation as a consequence of Abeta and pathological tau accumulation. Please put this in context with your above statement.
- Lines 205: You mix here cause and consequences: Not senescence contributes to ox stress and mito dysfunction but the other way around. Likewise, the latter 2 processes are connected and only mitochondrial ROS are able to generate DNA breaks while there are many more other mito dysfunctions unrelated to ROS. Please correct.
- Line 207: Please add “circle” after “vicious” since otherwise the sentence is incomplete.
- Line 211: It should be “MOUSE brains”, no “the” in front. Also, please remove “abilities” after “memory”.
- Lines 213-217 require references. In particular the statement that "dysfunctional autophagy" promotes GATA4 accumulation seems not correct. Rather GATA4 is a key regulator of the SASP and senescence and accumulates during cellular senescence. “Once the cell experiences senescence-inducing stimuli, the interaction between GATA4 and SQSTM1 decreases, and GATA4 escapes from autophagic inhibition and accumulates. This accumulated GATA4 initiates a transcriptional circuit to activate NFKB/NF-kB and the SASP” in response to DNA damage signaling. Mechanism: GATA4is normally degraded by p62-mediated selective autophagy, which is inhibited in senescent cells. See for example, DOI: 1080/15548627.2015.1121361.
- Line 227: ref. 41 is wrongly cited here and is a study about iron, not senolytics in mouse models. Please provide a correct reference or cite ref. 39 which used senolytics on a PS19 mouse model. Also: “improved deficits” a misleading term. Better say “improved cognitive abilities”.
- Line 229: Please replace “is” with “are” since astrocytes are plural.
- Line 231: requires a reference.
- Line 232: Please replace “a” with “the”.
- Line 237: Please remove “the”.
- Line 260: Please add “factor levels” after “SASP”.
- Table 1: Please clearly name the 2 columns and provide references for the table content. Do not use capital letters in column 1.
- 2nd column, row 1, line 3: Please replace “secreted” with “secreting” since otherwise grammar is wrong.
- 2nd column, row 2, line 2: Please replace “inducing” by “induce” since otherwise grammar is wrong.
- 2nd column, row 3, line 1: Please add “factors” after SASP
- 2nd column, row 4, lines 13-16: IGF-1, FGF2 and VEGF are NOT neurotrophins, please correct. Line 21: Please mention mitochondria as the organelle where ATP is generated mainly.
- 2nd column, row 5, line 3: Please remove “the” in front of “tau”.
- Line 248: Protein aggregates are not just transported but engulfed into autophagosomal membranes that form vesicles: autophagosomes that then lyse with lysosomes and their membranes.
- Line 251: Please replace the unsuitable term “escalation” with a more suitable ones such as “accumulation”.
- Line 255: Please provide a reference.
- Line 260: Please separate “lysosomal” and “disorders”.
- Lines 261 and 264 require references.
- Line 263: Please remove “5’” since it does not belong to the name of the kinase. Moreover, energy status and AMP/ATP ratio are synonyms and thus, no “respectively” is required. Also, mTORC1 is also a sensor for nutrient levels.
- Line 264: Please replace “The two of them” with “Both”.
- Line 265: When explaining the abbreviation of ULK1, you cannot say “also known as” since it is just an abbreviation.
- Line 268 requires a reference.
- Line 281: please replace “The” with “A” in front of summary. Also, it should be “it’s impact ON AD...”. Correct also in the figure title.
- Figure 1 requires a proper figure legend, not just a name. For example, what does the arrow from AMPK to mTORC1 mean? Please explain properly the processes shown in figure 1 independent of the text. What are the arrows in the black icons mean? Importantly, the scheme just shows a completely general process of autophagy (not comprehensive since many additional factors such as p62 etc. are missing). Where is the reference to AD here? Please improve the figure.
- Line 286: What do you mean with “following disease”? Your topic is AD, so please specify. Importantly, grammar of the sentence is completely wrong. Please rephrase and correct.
- Lines 288-290: Mitophagy is a specific selective type of autophagy where the substrate are defective mitochondria.
- Line 295: Please replace “failure” which would lead to immediate cell death with a more gradual term such as “decline”, “impairment” etc.
- Lines 298 and 300: “Division” of mitochondria is term “FISSION”. Please replace.
- Line 299: RNAs are NOT expressed, ONLY genes are. “level” is sufficient, remove “expression”.
- Lines 300-301: Please describe the conditions under which mitophagy is disrupted since fragmented mitos as described in the previous sentence, are prime substrates for mitophagy. Please specify and clarify and keep logics between your statements.
- Line 308: Please remove “expression” since proteins are NOT expressed, ONLY genes are. Also, a reference is required.
- Line 312: Reference 52 is a general review on mitophagy in NDDs and not a specific study on nematodes. Please cite original studies specifically and not just other reviews citing specific studies.
- Line 314: It should be “proven”, but better is “shown/demonstrated”
- Line 316: You have not described any biogenesis of mitos previously.
- Your review lacks a description of senescent neurons although there are thousands of papers on that topic, see for example: doi: 10.3390/ijms23041989., doi: 10.3390/ijms23084351. And many more
- Lines 319-22: require a reference.
- Line 329: AS already mentioned above, these 2 growth factors are general ones and not neurotrophins or brain-specific. Please correct.
- Line 330: What is activated and produces ROS? Please specify since it is unclear since you talked about growth factors and cytokines in the previous sentence, so no logical flow exists here. Moreover, what type of neuronal cell you refer to here?
- Line 332: Where from are cathepsins released? Your sentences are sloppy, please write scientifically clear and concise sentences.
- Line 333: here you refer to neuronal apoptosis but you have not described any interaction between microglia and neurons for the reader yet.
- Line 339: please replace “were” with “are” since these processes still occur.
- Line 341: Similarly to point 95: Now you refer to astrocytes that promote neuronal apoptosis without having described any interaction between these cell types yet.
- Also, please provide more details on molecular pathways for apoptosis induction and justify that you describe these processes under the heading “senescent microglia” since you have not described any senescence-realted process here yet.
- Line 349: Please provide a reference/references when referring to “studies”.
- Line 354: Please give more details on “Abeta phagocytosis”.
- Line 356: Please replace “aging” which refers to whole organisms by “senescent” which is the first and only mention of this topic within the wole section. Please also specify the relationship between “dystrophic” and “senescent”.
- Line 360: please add a comma after “described”.
- Line 361: The bystander that you describe does not destroy any cells (it is an unscientific term!), but induce senescence in neighbouring cells. Please correct.
- Line 362: the term "toxic senescent cells" does not exist and is thus highly unscientific.
- Line 365: Please replace “is” with “are” since plural is required. Also, Exercise” is a term mainly used in singular as a general intervention independent on single “exercises” which are all contained in the summary term.
- Line 367: Please give references for your statement.
- It is unclear whether all the described interventions have also been shown on senescent cells in brain which certainly exist but you do not cite a single such age-related intervention study on brain, like, for example: doi: 10.1111/acel.13296. and other papers from that group. Importantly, you also fail to make any references to AD here.
- Line 370: Please replace “Next” which is wrong grammar with “ANOTHER”.
- Line 381: Please provide original references for these studies instead of just citing previous reviews.
- Lines 381-3: Why do you talk about adipocytes here when your topic is brain and NDDS such as AD? Please provide specific studies which exist, (see above) and you describe in the below text.
- Lines 394/5: Please clarify what CKII is.
- Line 396: a review can not have a discussion since it lacks original data. Moreover, your conclusions are very brief anyway, so please remove “discussion”.
- Line 402: Please specify what you mean with “decreased regenerative ability” on a cellular level which in my view does not exist and the term is rather used for tissues than single cells. Also, what is “proper apoptosis”? Is there any “improper one”?
- Line 404: In your conclusion you now mention neurons under the topic of senescence without having described it in your review (see above) which needs to be added.
- Line 404: What ere “aging forms” that were found in brain cells? Please use a proper scientific terminology.
- Lines 405, 408, 412 and 415: Please replace “aging” with “senescence”, see explanation above.
- Line 408: Please replace “a
- Lines 418: AD is a NDD, not just a concept. Please amend and correct.
Language is more or less ok, but needs some adjustments which I mention to the authors in my detailed comments.
Round 2
Reviewer 2 Report
Comments and Suggestions for Authors
The authors addressed the majority of my comments and added a section about neurons which was missing before as the most important cell type for AD and NDDs, but there are still shortcomings with incorrect statements as well as clumsy and incorrect language/grammar. In most cases, the authors just deleted statements instead of rectifying it or claimed to have changed mistakes while they did not. For example, they still did not add a figure legend to the only figure while claiming it in the response letter. Moreover, there is still nothing specific for AD in this scheme although it is mentioned in the title. The many mistakes still show the lack of knowledge from the authors in the field of senescence and cellular processes. Please see details below:
- Line 22: You still have it wrong: a PHENOTYPE CANNOT be secreted! Just its FACTORS!
- Line 23: “affects variously” is not good English. Please rephrase, perhaps use “differently” instead.
- Line 71: Please remove “the” in front of Pubmed.
- Line 75: Please remove “a brief reminder of” since it sounds rather unscientific. Just briefly introduce the disease as you do withour “reminding” anybody of anything.
- Lines 81/2: estimated TO BE around...” is future and contradicts the year 2020 which is in the PAST! There should be figures for that year, alternatively use prognosis for the future which is beyond 2025!
- Line 86: remove “the” in front of APP...
- Line 175: What do you mean with “have their homogeneity”, firstly, language is clumsy and what do you mean with “homogeneity”? Cells are always heterogeneic and it should be related to certain parameters. Please clarify.
- Line 176: please add “the” in front of “nucleus...”.
- Line 198: Senescence is not cause dby aging, but the other way around! Also, what do you mean with “improper removal”? Apoptosis is an alternative mechanism to senescence and does not cause any senescence! Please correct this false scientific content.
- Line 208: senescence is not related to “changes in the DDR” but BY the DDR! Please correct.
- Line 212: SASP components can both induce and inhibit apoptosis, depending on the context and cell type. Please correct.
- Lines 212/3: You cannot transmit senescence with senescence! The process is called “bystander effect” and means that sernescencet cells secrete SASP factors which render neighbouring cells also senescent in a paracrine fashion. Please correct.
- Line 215: what is “TauO”? Please explain unless it is a typo.
- Table 1: Instead of giving references in the title you should give it for each statement given in the right column.
- 1st row, 2nd column: “secreting SASP” is incorrect. Please add “factors”.
- Row 4, 2nd column on page 7: You still list IGF and VEGF as neurotrophins which is incorrect. They are just general growth factors. Please correct. I thought I had already criticised that in the first review.
- Line “it” referring to neurons is wrong language, it should be “They”.
- Line 267: Do you refer to genes or proteins? In the latter case please remove/replace “expression”.
- Line 274: Please only list the name of the FIRST author, not several.
- Line 277: Here you clearly refer to proteins, so please remove/replace “expression”!
- Line 279: The “late” indicator is not logical relating to the EARLY occurrence mentioned before. Please clarify or remove.
- Line 280: “A persistent...DDR”.
- Line 290: How can cultured neurons induce astrogliosis? Are these mixed cultures or how was the experimental setting? Please clarify and specify details of described conditions. Does this also belong to ref. 57? Then cite it earlier or give the reference.
- Line 294: As above: Cite ONLY the FIRST authors surname without initials.
- Line 361: The statement is wrong since there are also proteasomes that degrade specific types of misfolded proteins. Please add and describe the process which is as important for proteostasis in the brain as autophagy.
- Figure 1: There is still NO figure legend although you claim to have added it in your responses. Importantly, while you added AD in the figure heading, there is nothing specific for AD in this scheme! Please make sure that it relates to AD SPECIFICALLY.
- Line 461: Please add “positive” between “p16INK4A” and “senescent cells”, also: “with A ... cocktail OF...”, otherwise language is incorrect.
- Line 464: You should use past tense “HAD improved” and also add the condition” under this treatment/under these conditions/in this experiment etc”, otherwise the sentence has no relation to the above sentences.
- Line 483: Please replace “form” with a more suitable term, best to use “cells”.
Language is still rather clumsy and I mentioned the remaining most crude mistakes to the authors, but it should still be improved, best by a native speaker.
